# The Relationship between Predominant Polarity, Lifetime Comorbid Anxiety Disorders and Subjective Quality of Life among Individuals with Bipolar Disorder in Singapore

**DOI:** 10.3390/ijerph20021155

**Published:** 2023-01-09

**Authors:** Savita Gunasekaran, Wen Lin Teh, Jianlin Liu, Laxman Cetty, Yee Ming Mok, Mythily Subramaniam

**Affiliations:** 1Research Division, Institute of Mental Health, Singapore 539747, Singapore; 2Department of Mood & Anxiety, Institute of Mental Health, Singapore 539747, Singapore

**Keywords:** bipolar disorder, comorbid anxiety, predominant polarity, quality of life, Singapore, Asia

## Abstract

Background: Depressive features and comorbid anxiety disorders are two discrete but interconnected clinical features that have been reported to be associated with a poorer quality of life (QoL) among individuals with bipolar disorders. However, the relationship between manic features and quality of life is less conclusive. The present study aimed to assess differences in QoL among bipolar outpatients who present with either depressive predominant polarity (DPP), manic predominant polarity (MPP) and/or a lifetime diagnosis of comorbid anxiety disorders in Singapore. Methods: Data from 74 outpatients in Singapore diagnosed with bipolar disorder were collected. Sociodemographic information, the polarity of most episodes (2 out of 3), the diagnosis of anxiety disorders and QoL were obtained from a self-reported interview and/or through clinical records. QoL was measured using the abbreviated version of the World Health Organization questionnaire. We used multivariate regression models to determine the relationships between predominant polarity, lifetime comorbid anxiety disorders and QoL in physical health, psychological health, social relationships and environment domains. Results: After adjusting for covariates, individuals with DPP scored poorer for WHOQOL-BREF for all four domains as compared with individuals with indeterminate polarity. As compared to individuals with indeterminate polarity, individuals with MPP scored poorer for WHOQOL-BREF social relationships. Lastly, individuals with lifetime comorbid anxiety disorders scored poorer for WHOQOL-BREF physical health, social relationships and environment. Discussion and Conclusions: The present study provides preliminary support for the relationship between DPP, lifetime comorbid anxiety disorders and poorer QoL, paving the pathway for future research with larger samples to utilise our study design to verify our results.

## 1. Introduction

In recent decades, the term quality of life has been increasingly used in mental health recovery literature, with the goal of focusing on patients’ subjective experiences and perceptions. Quality of life (QoL) is a multidimensional construct that measures the degree of well-being that an individual experiences [1]. Although there is no standard definition for this tangible concept, there is a general consensus that quality of life consists of various aspects of a person’s life such as emotional and physical health and social and occupational well-being [1]. This shift towards focusing on quality of life emphasises not only symptom reduction, but also maintaining a meaningful existence and autonomy among individuals with mental illnesses [1]. Poor QoL is often a cause or consequence of many mental illnesses, although the extent of the impact on QoL varies [2].

Bipolar disorders are described by the Diagnostic and Statistical Manual of Mental Disorders 5th edition (DSM-5) as periods of at least one week where a person has an elevated or irritable mood or a hypomanic episode that lasts at least four days in a row with major depressive episodes [3]. The disorder impacts different domains of an individual’s life such as education, relationships and occupational productivity [4]. The extant literature has suggested a pattern of poor perceptions of QoL among bipolar patients, compared to other largely studied psychiatric disorders [5]. Singapore is a developed Southeast Asian country with a population of approximately five million people [6]. The estimated lifetime prevalence of bipolar disorder is 1.6% [6]. In the local literature, studies have investigated the prevalence and epidemiology of the illness through nationwide studies [7,8], such as a 2016 study that revealed a concerning increase in the proportion of individuals with bipolar disorder showing severe manic or depressive symptoms as compared to five years ago [8]. This shift creates a growing need to understand both manic and depressive symptomology separately.

There is increasing evidence that predominant polarity has implications for therapeutic and prognostic outcomes [9]. Predominant polarity identifies patients who have episodes of primarily depression and/or mania—if at least two-thirds of lifetime episodes of an individual with bipolar disorder are decompensated to one pole of the illness, that pole is defined as the predominant polarity [10,11]. Manic predominant polarity (i.e., MPP) is characterised by more frequent episodes of manic characteristics in general [12]. Individuals who are usually decompensated to the depressed pole (i.e., depressed predominant polarity; DPP) have more frequent depressive episodes [12,13]. The associations between predominant polarity and variables indicative of severity and outcomes of bipolar disorders have been captured in the literature [14]. For example, studies have revealed that DPP is associated with a poorer prognosis, worse response to treatment [9,15], more suicide attempts, more relapses, prolonged episodes and delayed diagnosis of bipolar disorders [14,16]. MPP has been associated with a higher prevalence of psychotic symptoms, a greater number of hospitalisations, better responses to mood stabilisers and poorer performances in various cognitive domains [14,17,18,19].

There is extant literature highlighting the association between depressive symptoms and comorbid anxiety, especially in its relationship with QoL. Specific to predominance, a reported difference between DPP and MPP is that individuals with the former are more likely to have a comorbid anxiety disorder [12]. It is well reported in the bipolar literature that depressive symptoms are inversely correlated with QoL scores [4,20], and that depressive symptomology and comorbid anxiety disorders are two discrete but interconnected variables that negatively impact the quality of life among this population. In a study conducted by Kauer-Sant’Anna et al. (2007), having a diagnosis of anxiety disorders (dichotomously measured) was correlated with reduced perceptions of quality of life in psychological health and social relationships [21]. However, after including the level of depressive symptoms in the model, having a diagnosis of anxiety disorders only significantly influenced QoL in psychological health [21]. Whilst the relationship between depressive symptoms and comorbid anxiety disorders and their influence on QoL has been explored, the relationship between comorbid anxiety disorders and DPP and their influence on QoL have yet to be explored. Exploring this would be beneficial in exploring if comorbid anxiety disorders are associated with lower levels of QoL above and beyond frequent episodes of depression.

One of the current issues with understanding the quality of life among individuals with bipolar disorders is that most existing studies focus on the disorder as a whole, lacking generalisability to individuals with differing symptom decompensation. The differences in the prognosis and treatment outcomes for both DPP and MPP have been delved into in the current literature [15,22,23]; however, quality of life has yet to be explored in this context. This is indeed surprising considering that quality of life has continuously been emphasised as an important factor in bipolar recovery and prognosis, especially in recent years. Considering the complexity of the recovery process for psychiatric illnesses such as bipolar disorders, the possibility of a “cure” is difficult, and thus clinicians have shifted their focus to improving patients’ quality of life. Additionally, the concept of quality of life is largely used in the context of assessing treatment effectiveness, with the main goal of understanding if specific treatment approaches improve patients’ subjective well-being beyond alleviating symptoms [1].

This presents the question regarding the relationship between predominant polarity and quality of life above and beyond sociodemographic variables and other clinical correlates—if treatment and prognostic outcomes are different for each polarity, one would expect the quality of life of patients to differ for manic and depressive predominance. This research question is important for mental health practitioners working closely in the management of bipolar disorders in Singapore. The present paper aimed to investigate (1) the association between predominant polarity and subjective perceptions of quality of life after controlling for sociodemographic and clinical covariates, and (2) the association between comorbid anxiety disorders and quality of life after controlling for sociodemographic and clinical covariates.

## 2. Methods

### 2.1. Sample

A total of 100 outpatients from the Institute of Mental Health (IMH) who were diagnosed with bipolar disorders were invited to participate in this study from July 2020 to July 2021. The IMH is the sole tertiary care psychiatric hospital in Singapore that serves a patient population with bipolar disorders. The study team emphasised that participation is voluntary and nonparticipation will not affect standard clinical care or the relationship with the clinician or institution. The inclusion criteria were: patients with a DSM-IV diagnosis of bipolar disorder (I or II or NOS), the ability to understand English and between the ages of 21 to 65. This study excluded patients who were unable to understand English, were diagnosed with an intellectual disability and/or were cognitively impaired. The final sample included 74 participants.

### 2.2. Procedure

Participants underwent an interview with a trained research staff member utilizing a structured questionnaire. The interview included information on their quality of life, sociodemographic characteristics and clinical and course-of-illness variables. Ethical approval was received from the National Healthcare Group’s Domain-Specific Review Board (DSRB Ref No.: 2019/1082). Written informed consent was obtained from all the participants before the interview.

### 2.3. Materials

Sociodemographic, clinical and course-of-illness variables: Participants were interviewed on their age, gender, ethnicity, marital status, employment, education status, age of bipolar onset, diagnosis types, history of psychotic features during and outside an episode, previous suicide attempts, family history of mood disorders (first-degree relative), predominant polarity and self-reported lifetime comorbid anxiety disorders (i.e., “Have you ever had an anxiety disorder?”) among other variables. Predominant polarity (i.e., “What is the polarity of most of your episodes (approximately 2 out of 3 times)?”) was documented as a 5-level categorical variable comprising depression predominant polarity, manic predominant polarity, hypomanic predominant polarity, no predominant polarity or less than three lifetime episodes. We collapsed the variable to a 3-level categorical variable per the Barcelona proposal: according to its definition of predominant polarity, individuals are required to present at least two-thirds of lifetime episodes to be decompensated to either pole to be included into either MPP or DPP categories [24]. If neither of the polarity had a two-thirds majority, we categorised these participants into indeterminate predominant polarity [24]. This categorisation of predominant polarity has been widely used in many recent studies [14,25]. When participants were unsure, self-reported variables of interest were verified with the attending clinician at the time of the interview and/or through medical records to minimise recall bias.

Abbreviated World Health Organization Quality of Life questionnaire (WHOQOL-BREF, 1996): Quality of life was measured using the WHOQOL-BREF, a 26-item multidimensional scale that encompasses four domains (physical health, psychological health, social relationships and environment). The instrument is sensitive to subjective quality of life assessments in clinical populations of severe and long-term psychiatric illnesses [26]. The WHOQOL-BREF offers a concise, valid and reliable alternative to the WHOQOL-100 [26]. Additionally, this measure has been locally tested for validity and reliability, eliciting good internal consistency, test–retest reliability and validity [27]. In this study, Cronbach’s alpha was calculated to determine the internal consistency of the instrument in our sample. The values ranged from moderate (0.6 for the social component) to good (0.7 for the physical component and 0.8 for the psychological and environmental components) in this study. Items were rated from 1 to 5 on a Likert scale, with lower scores indicating poorer, more negative perceptions of QoL [28].

### 2.4. Data Analysis 

The descriptive analysis included counts and proportions for categorical variables. We calculated means and standard deviations for the continuous variables (i.e., age; age of onset). To analyse the associations between predominant polarity and lifetime comorbid anxiety disorders with WHOQOL-BREF, multivariate regression analyses were conducted. Four multivariate regression models were analysed with each WHOQOL-BREF domain as the dependent variable. Lifetime comorbid anxiety disorders and predominant polarity were added to the model, controlling for sociodemographic factors—i.e., age, sex, ethnicity, education level, employment status and age of onset of illness [4,24]. Missing variables for the WHOQOL-BREF were dealt with according to the manual. All other missing variables were deleted listwise. All assumptions of linear regression were not severely violated. For statistical inference, 95% confidence interval (CI) parameter estimates and Cohen’s *f*^2^ effect sizes were calculated. When the upper and lower CIs do not cross zero, the effects are interpreted as significant. Effect sizes were interpreted according to Cohen’s guidelines for multiple regressions (small = *f*^2^ ≥ 0.02, medium = *f*^2^ ≥ 0.15, large = *f*^2^ ≥ 0.35) [29]. The current study utilised SPSS 24 for Windows, with an alpha value of 0.05 utilised for all models.

## 3. Results

Table 1 presents the descriptive statistics for the demographic, clinical and course-of-illness variables of the subset of eligible participants (*n* = 74). A total of 31 participants (41.3%) reported having MPP compared to the 22 participants (29.3%) who reported having DPP. A total of 14 (18.7%) participants had a lifetime diagnosis of comorbid anxiety disorders. Among the participants with comorbid anxiety, six (8%) participants presented with DPP and six (8%) participants presented with MPP.

### 3.1. Predominant Polarity and WHOQOL-BREF Domain-Specific Scores

As compared to individuals with indeterminate predominant polarity, those with DPP had lower scores in the WHOQOL-BREF physical health (β = −2.96, 95% CI −5.55 to −0.37, *f*^2^ = 0.20), psychological health (β = −3.42, 95% CI −5.98 to −0.86, *f*^2^ = 0.27) and environmental health (β = 3.05, 95% CI −5.49 to −0.61, *f*^2^ = 0.23). Additionally, those with MPP had a lower score in the WHOQOL-BREF social relationships (β = 3.45, 95% CI = −6.15 to −0.74, *f*^2^ =0.24) as compared to individuals with indeterminate predominant polarity.

### 3.2. Lifetime Comorbid Anxiety Disorders and WHOQOL-BREF Domain-Specific Scores

Furthermore, having lifetime comorbid anxiety disorders was associated with lower WHOQOL-BREF scores for physical health (β = −2.91, 95% CI = −5.47 to −0.36, *f*^2^ = 0.20) as compared to not having lifetime comorbid anxiety disorders. There was no other significant difference between participants with lifetime comorbid anxiety disorders and those without. Besides the variables of interest, all the control variables were not significant in the multivariable model. Table 2 presents the summary of the general linear model parameter estimates for the variables of interest.

## 4. Discussion

The present research aimed to investigate the relationship between predominant polarity, lifetime comorbid anxiety disorders and QoL in physical health, psychological health, social relationships and environment domains. When compared with individuals with indeterminate predominance, the results of this study suggest that individuals with DPP have poorer perceptions of QoL in the physical, psychological and environment domains, but share no significant difference in the social relationships domain. The present study found similar effect sizes for DPP on physical, psychological and environment domains; medium effect sizes were generated for each of the domains, which may suggest that depressive symptoms negatively influence these domains as a whole as a sample.

Secondly, the results revealed that MPP was moderately associated with lower WHOQOL-BREF scores in the social relationships domain. Due to the novelty of examining the relationship between predominant polarity and QoL, not much research has been previously conducted investigating why manic predominance could be only associated with poorer social relationships. However, some speculations can be made—firstly, Gazelle and colleagues have reported that higher scores of manic symptoms were associated with significantly lower QoL scores in the social relationships domain, a finding that is analogous to the present paper [30]. Additionally, according to Siegel and colleagues, symptoms during manic episodes, e.g., increased irritability, risky behaviours and hyperverbal speech, etc. can possibly jeopardise interpersonal relationships with close ones and consequently might affect one’s perception of the quality of their personal relationships [31]. Similarly, a qualitative study with bipolar patients conducted by Owen et al. (2017) revealed that patients opined that risk-taking and disinhibited behaviours during these episodes can lead to the breakdown of important relationships [32]. Whilst the aforementioned studies did not directly measure manic predominance, frequent manic episodes are a core feature of MPP, suggesting that the repeated lifetime episodes of mania could possibly lead to such behaviours that inhibit the quality of one’s social relationships. This suggests a focus on improving social relationships among patients who present with this predominance.

Additionally, the lack of a relationship between DPP and lower scores of WHOQOL-BREF social relationships can be attributed to cultural factors such as the structure of the family in Asia as compared to Western counterparts [33]. Furthermore, social acceptance and support given to individuals with MPP and DPP differ across both groups. For one, individuals who present with manic episodes are less likely to be accepted by their close ones as compared to individuals who present with depressive episodes [34]. Furthermore, people felt more fear, irritability and lesser concern towards individuals with manic episodes compared to those with depressive episodes [34]. There is some evidence pointing towards lesser support and social acceptance towards individuals with manic episodes as compared to those with depressive episodes, explaining why only MPP, characterised by repeated manic episodes, was attributed to lower WHOQOL-BREF social relationships scores.

The results also revealed that having lifetime comorbid anxiety disorders was associated with poorer WHOQOL-BREF scores in the physical health domain. Besides physical health, individuals with a lifetime diagnosis of comorbid anxiety disorders did not significantly differ in their perceptions of QoL as compared to individuals without. The results of the study imply that the association of comorbid anxiety disorders with WHOQOL-BREF physical health was kept, even with DPP serving as a confounding factor in the model. Physical health was the only domain that was independently associated with both a lifetime diagnosis of anxiety disorders and DPP. This relationship could be because of several postulated reasons—comorbidity in bipolar disorders, especially anxiety comorbidity, has been linked to aspects of WHOQOL-BREF physical health such as poorer subjective sleep quality, a worse ability to perform daily activities (i.e., poorer global assessment of functioning scores) and minimal employment [35,36]. Furthermore, studies have shown that symptoms of comorbidity and depression have been frequently relieved through self-medication, thus increasing patient health issues [37].

Additionally, using a universally utilised structured measure of QoL (i.e., WHOQOL-BREF) in the present paper provides us with insights into the current sample of bipolar outpatients’ perceptions of QoL in comparison to other samples of not only bipolar patients but also patients with other psychiatric disorders in Singapore. In recent years, Teh et al. (2021) conducted a cross-sectional study investigating the differences in WHOQOL-BREF scores among individuals with schizophrenia, other nonaffective psychiatric disorders and affective psychotic disorders [38]. In comparison to the sample of schizophrenia patients in Teh and colleagues’ study (*n* = 231), the present sample of bipolar outpatients scored higher in all WHOQOL-BREF domains. However, in comparison to other nonaffective psychotic disorders (excluding schizophrenia) (*n* = 109), the present sample only scored higher in the environment domain. Additionally, Teh and colleagues categorised individuals with either Bipolar I disorder or Major Depressive Disorder and with psychotic features into an overarching category of individuals with affective psychotic disorders [38]. Compared to this categorised sample of individuals with affective psychotic disorders (*n* = 25), the present sample of individuals with bipolar disorder scored similarly in QoL scores across all domains.

The present research sheds light on the path to follow regarding future research into QoL and bipolar disorders in Singapore. The negative correlations between predominant polarity, comorbid anxiety and WHOQOL-BREF scores suggest that if similar results are found in future studies employing a longitudinal analysis with larger samples, it has implications for mental health service delivery for bipolar patients in Singapore beyond primary mental health care. The present study encourages more research into this topic, with predominant polarity emerging as a promising clinical specifier associated with QoL scores. Previous studies have used other clinical correlates such as symptom severity as predictors of QoL [20]. The use of such specifiers alongside the more novel predominant polarity in future studies can provide insights into the magnitude of influence of such variables on QoL scores. This would in turn provide valuable information on what clinical specifiers and symptoms to pay close attention to with regard to holistic treatments targeted at improving QoL.

Lastly, the results of the study must be interpreted in view of some limitations. The cross-sectional nature of the study did not allow for the analysis of dynamic interactions between QoL and the variables of interest, and therefore, any possibility of bidirectional effects could not be investigated [39]. QoL is a multidimensional concept, and we were unable to adjust for potential confounders that were not collected in the study, such as the number of depressive or manic episodes and symptom severity [23]. Moreover, the small sample size might hamper some detection of significant associations. Some retroactively measured variables (e.g., self-reported diagnosis of anxiety) might also be subjected to recall bias.

## 5. Conclusions

Despite these limitations, the present study has strengths. It is the first study that examines the relationship between quality of life among individuals with bipolar disorders and predominant polarity using the WHOQOL-BREF in Singapore. Identifying the factors that are largely correlated with QOL is essential so that they can be targeted in treatments. Our results indicated that depressive predominance, manic predominance and lifetime comorbid anxiety disorders are independently associated with different components of poor QoL. Future research can rely on our findings to design studies with a larger number of participants and verify our results.

## Figures and Tables

**Table 1 ijerph-20-01155-t001:** Summary of sociodemographic, clinical and course-of-illness variables.

Variable	
Age, mean (SD)	38.29 (12.50)
Gender, *n* (%)	
Male	34 (45.9)
Female	40 (54.1)
Ethnicity *n* (%)	
Chinese	57 (77.3)
Malay	8 (10.8)
Indian	6 (8.1)
Others	3 (4.1)
Marital Status, *n* (%)	
Single	36 (48.6)
Married	26 (36.1)
Divorced/Separated	12 (16.2)
Education level, *n* (%)	
Secondary and below	16 (21.9)
Pre-tertiary	20 (27.4)
Degrees and professional qualifications	37 (49.3)
Employment status, *n* (%)	
Currently not employed	26 (34.7)
Employed (full/part-time)	48 (64.0)
Age of onset, mean (SD)	26.09 (10.25)
Diagnosis type, *n* (%)	
Bipolar I	67 (90.5%)
Bipolar II	6 (8.1%)
NOS	1 (1.4%)
History of psychotic features during episode, *n* (%)	48 (64.9%)
History of psychotic features outside episode, *n* (%)	7 (9.5%)
Previous suicide attempt, *n* (%)	33 (44.6%)
Family history of mood disorder *, *n* (%)	19 (25.7%)
Predominant polarity (2 out of 3 episodes), *n* (%)	
Depressive predominant polarity (DPP)	22 (29.3)
Manic predominant polarity (MPP)	31 (41.3)
Indeterminate predominant polarity	20 (26.7)
Comorbid anxiety, *n* (%)	14 (18.7)
WHOQOL-BREF, mean (SD), *n* = 74	
Physical health	14.3 (2.5)
Psychological health	13.0 (2.8)
Social relationships	13.8 (2.8)
Environment	14.8 (2.3)

* Family history of mood disorder in first-degree relative.

**Table 2 ijerph-20-01155-t002:** Summary of general linear model parameter estimates for WHOQOL-BREF scores for physical health, psychological health, social relationships and environment.

	Quality of Life (WHOQOL-BREF)											
	Physical Health			Psychological Health			Social Relationships			Environmental Health		
*r* ^2^	0.56			0.58			0.49			0.45		
	β (SE)	95% CI	Cohen’s *f*^2^	β (SE)	95% CI	Cohen’s *f*^2^	β (SE)	95% CI	Cohen’s *f*^2^	β (SE)	95% CI	Cohen’s *f*^2^
**Predominant Polarity**												
Indeterminate ^a^	Ref											
Manic predominant polarity	−0.46 (1.16)	[−2.84, 1.91]	0.01	−0.59 (1.14)	[−2.93, 1.76]	0.01	−3.45 (1.32)	**[−6.15. −0.74]**	**0.24**	−2.06 (1.09)	[−4.30, 0.17]	0.13
Depression predominant polarity	−2.96 (1.26)	**[−5.55, −0.37]**	**0.20**	−3.42 (1.25)	**[−5.98, −0.86]**	**0.27**	−2.89 (1.44)	[−5.84, 0.06]	0.14	−3.05(1.19)	**[−5.49, −0.61]**	**0.23**
**Lifetime diagnosis of anxiety disorders**												
No	Ref											
Yes	−2.91 (1.25)	**[−5.47, −0.36]**	**0.20**	−2.08 (1.23)	[−4.60, 0.44]	0.10	−2.21 (1.42)	[−5.12, 0.70]	0.09	−1.56 (0.76)	[−3.08, −0.04]	0.08

Note. Analyses involved regressing quality of life domains on both predominant polarity (manic predominant polarity, depression predominant polarity and neither) and comorbid anxiety disorders (yes or no) while controlling for sociodemographic information such as gender, ethnicity, marital status, highest education levels, employment status, age, and clinical variables such as age of bipolar onset, diagnosis type, history of psychotic features during an episode, history of psychotic features outside an episode, previous suicide attempts and family history of mood disorders (first degree relative). Values in bold indicate significance because 95% upper and lower confidence intervals for the variables did not cross zero. ^a^ Indeterminate refers to participants diagnosed with bipolar disorders who were not diagnosed with either manic predominant polarity or depressive predominant polarity.

## Data Availability

The authors’ government law and institution only permits sharing of human participant data with researchers with whom they have a written agreement. These restrictions have been imposed by our Institutional Review Board (IRB) and Institutional Committee (NHG Domain Specific Review Board and IMH Clinical Research Committee). Our IRB guidelines suggest that a Research Collaboration Agreement (RCA) be signed with collaborating parties. However, data sharing with clear research purposes are available upon request at this contact: Research Director of IMH, Associate Professor Mythily Subramaniam (mythily@imh.com.sg).

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
