# Peer review of "The Relationship between Predominant Polarity, Lifetime Comorbid Anxiety Disorders and Subjective Quality of Life among Individuals with Bipolar Disorder in Singapore"

_ijerph, 2023, doi:10.3390/ijerph20021155_

Round 1

Reviewer 1 Report

First of all, I would like to congratulate you on your work, especially given the difficulty of studying the subject you propose.

Summary

In general it explains well what the study is about, but I think that the principle of the predominantly manic polarity term could also be included, as only depression and anxiety are mentioned.

Introduction

The background and the state of the question are clearly stated, but I have doubts about some of the studies they cite, although some are very old, in others the sample they use are participants who already have other mental health problems and it is not very clear to me if this can be put as a background or if they should look more into other studies that work on predominant polarity. Also, I am missing a bit of a connection between the different associated disorders such as depression, anxiety and mania. This is a matter of writing it up in a more threaded way that can focus the reader in his or her study.

Methods

I am concerned about the sample, I think 75 participants is too few to draw meaningful conclusions, and it is a cross-sectional study that does not take into account external factors that may alter the results. I would encourage you to continue to collect data over several years and then analyse it in a way that makes your results more robust. With this sample, it is not possible to draw true conclusions. In addition to the socio-demographic variables, they should also include medical and/or psychological variables to find out what stage each participant is at in order to unify the sample.

As for the design, I think it is adequate for a cross-sectional study, but as I said, it seems too small to draw conclusions. In fact, in your sample you only have 22 participants who suffer from predominant polarity, which is the main object of study. As for the analysis, I think it is correct but I miss the use of some other validated and consolidated instrument to analyse the results, since only the WHO health instrument is very simple and not very verifiable. I would advise you to redo the analysis using other instruments to validate your data.

Discussion

The discussion is correct, the problem is that the studies with which you compare your results do not seem to have followed the same steps as yours and therefore I do not believe that they can be compared and make the claims you make. Perhaps it is because of the difficulty of measuring the variables you raise and the fact that there is no standardised way of doing so, which makes comparisons very difficult.

Reviewer 2 Report

Thank you for the opportunity to review this interesting and timely manuscript entitled “The relationship between predominant polarity, lifetime comorbid anxiety disorders and quality of life among individuals with bipolar disorder in Singapore” submitted for publication in International Journal of Environmental Research and Public Health. I consider the manuscript as a valuable and underlying work. I did really enjoy reading this well-written manuscript and have no comments. I do accept the manuscript with a very minor revision in the discussion.

The study investigated the relationships between several quality of life domains and two clinical variables – predominant polarity and lifetime comorbid anxiety disorders. They used the WHOQOL-BREF for the first time. The results are helpful for future researchers in the field. Using different instrument can be considered as an added value to existing materials. The methodology was very clearly written and presented all the required information. The discussion is very well written addressing the main research question with discussing the evidence and arguments in the literature. Authors used appropriate references to address the topic.

The following sentence in the very last paragraph needs to be reworded. It more implies that the WHOQOL-BREF used to measure predominant polarity and comorbid anxiety disorders:

“It is the first study to use 279 the WHOQOL-BREF to examine predominant polarity and comorbid anxiety disorders 280 among patients with bipolar disorder in Singapore.”

Reviewer 3 Report

Dear Author(s):

I had the opportunity to review your paper entitled "The relationship between predominant polarity, lifetime comorbid anxiety disorders and quality of life among individuals with bipolar disorders in Singapore" submitted to the International Journal of Environmental Research and Public Health. In this survey-based quantitative study of 75 outpatients you found that all investigated disorders affect most of the reported self-perceived measures of quality of life. Overall, the study is well written and the methods mostly fine. However, all published studies should embody some “lessons learned” that I struggle to identify in your paper. Hopefully, my remarks will help you to improve your paper’s focus on the most surprising findings. Best of luck!

1.     First and foremost, your paper is a decent read—congratulations! However, what's missing is a big why. Why do we need your study? As you already know, insufficient research on a topic or controversies or disagreements may not be strong-enough reasons to motivate a study. Try to motivate the study better along these lines: (a) What is the study’s raison d'être? What are the research questions, and how will the answers move practice and theory forward? What do we know already about the problem, and what do we not yet know, but urgently must learn? And most importantly, what are your contributions? You might formulate one or two most important takeaways in the study’s conclusions supported by the empirical analysis. I am still puzzled about what I learned reading your paper that I did not know before—unfortunately, not much due to the less intriguing findings.

2.     Although highly relevant to your topic, some core studies are missing in your list of references, e.g., Belizario et al. 2019, Carvalho & Vieta 2019, Pal 2019, Yoldi‐Negrete et al. 2019. Specifically in the discussion section, you may acknowledge some of these past studies. A thorough literature review may further help you to identify your study’s unique selling point (USP).        

3.     Methodologically, there are small inconsistencies and missing information: What type of factor analysis did you run to calculate your factor scores? In line 128 you mentioned 75 included participants, your DV WHOQOL in Table 1 suggests 74 participants. However, you mentioned listwise deletion of missing values in line 165. Is your final sample 75 or 74? Following line 120 that all participants were diagnosed either with bipolar disorders “and/or comorbid anxiety” (line 13), how can the category “neither” in Table 1 be bigger (N=20) than comorbid anxiety (N=14). This implies at least N=6 participants without DPP, MPP, and comorbid anxiety, correct? As far as I understand all data is self-reported, both IV and DV? This might explain some discrepancies between perceptions and external assessments. However, in all survey-based studies with IV and DV drawn from the same source (i.e., respondent), you need to test for common method bias, for example, by using a comprehensive marker variable technique (Podsakoff et al. 2010, Williams et al, 2010) to avoid spurious findings. Moreover, consider adding a table for your final multiple regression estimators, including overall assessments of the DVs’ variance explanations. Did you test for the underlying assumptions of testing linear relationships, e.g., multivariate normality, variance homogeneity? If not, are your findings robust to violations of these assumptions? Consider non-parametric bootstrapping in case of violations.     

4.     Your data collection started shortly after a global pandemic (Covid-19). Do you believe this exogenous shock might invalidate your findings since isolation and several policies associated with Covid could have severe influence on the participants responses, especially briefly after the outbreak. You might add some thoughts in your discussion.

5.     Minor issues: line 83: “comorbid anxiety disorder is essential because it serves as…” – alternatively “comorbid anxiety disorders” in plural. Please consider a final copy-editing to avoid typos.

In general, your paper calls for a more compelling message. All the very best!

References

Belizario, G. O., Junior, R. G. B., Salvini, R., Lafer, B., & da Silva Dias, R. (2019). Predominant polarity classification and associated clinical variables in bipolar disorder: A machine learning approach. Journal of affective disorders, 245, 279-282.

Carvalho, A. F., & Vieta, E. (Eds.). (2017). The treatment of bipolar disorder: integrative clinical strategies and future directions. Oxford University Press.

Pal, A. (2019). Predominant polarity in bipolar affective disorder: a scoping review of its relationship with clinical variables and its implications. Indian Journal of Psychological Medicine, 41(1), 9-17.

Podsakoff, P. M., MacKenzie, S. B., Lee, J. Y., & Podsakoff, N. P. (2003). Common method biases in behavioral research: a critical review of the literature and recommended remedies. Journal of applied psychology88(5), 879.

Williams, L. J., Hartman, N., & Cavazotte, F. (2010). Method variance and marker variables: A review and comprehensive CFA marker technique. Organizational research methods13(3), 477-514.

Yoldi‐Negrete, M., Fresán‐Orellana, A., Jiménez‐Tirado, M., Martínez‐Camarillo, S., Palacios‐Cruz, L., Vieta, E., ... & Camarena Medellín, B. (2021). Ten‐year course of treated bipolar I disorder: The role of polarity at onset. Brain and Behavior, 11(11), e2279.

Round 2

Reviewer 3 Report

Dear Author(s):

I reviewed your paper for a second time. Overall, I am pleased with the revision. Only a few inconsistencies are left. Please adjust all statistics of Table 1 to eligible participants only, N=74. You still erroneously report 34 male and 41 female participants, in sum 75. Please further avoid splitting minus-signs and follow-up values. This might suggest positive values of the upper bonds of the confidence intervals in Table 2 of the appendix and in the document, e.g., line 202. There is a keyboard shortcut preventing such undesired new lines (ctrl+shift+space). Please also use long dashes as minus signs instead of short dashes: - vs. –. I spotted two or three more punctuation issues. So please use a professional proofread before final resubmission. Best of luck!